# Mental Health Disorders and Coping Strategies in Healthcare Workers during the COVID-19 Pandemic: An Analytical Cross-Sectional Study in Southeastern Mexico

**DOI:** 10.3390/ijerph20054230

**Published:** 2023-02-27

**Authors:** Jesús Maximiliano Granados Villalpando, Guadalupe del Carmen Baeza Flores, Jorge Luis Ble Castillo, Karla del Socorro Celorio Méndez, Isela Esther Juárez Rojop, José Antonio Morales Contreras, Viridiana Olvera Hernández, Sergio Quiroz Gómez, Sergio de Jesús Romero Tapia, Jesús Arturo Ruíz Quiñones, Crystell Guadalupe Guzmán Priego

**Affiliations:** 1Cardiometabolism Laboratory, Research Center, Health Sciences Academic Division (DACS), Juarez Autonomous University of Tabasco (UJAT), Villahermosa 86040, Mexico; 2Metabolic Disease Biochemistry, Research Center, Health Sciences Academic Division (DACS), Juarez Autonomous University of Tabasco (UJAT), Villahermosa 86040, Mexico; 3Lipid Metabolism, Research Center, Health Sciences Academic Division (DACS), Juarez Autonomous University of Tabasco (UJAT), Villahermosa 86040, Mexico; 4Health Sciences Academic Division (DACS), Juarez Autonomous University of Tabasco (UJAT), Villahermosa 86040, Mexico; 5Research Center for Tropical and Emerging Diseases, High Specialty Regional Hospital “Juan Graham Casasús”, Villahermosa 86126, Mexico

**Keywords:** COVID-19, healthcare workers, mental health disorders, coping strategies, depression, anxiety, stress

## Abstract

Mental health disorders are relatively common in the general population and were already an important issue for the healthcare sector before COVID-19. COVID-19, being a worldwide crucial event and evidently a great stressor has increased both the prevalence and incidence of these. Therefore, it is evident that COVID-19 and mental health disorders are closely related. Moreover, several coping strategies exist to endure said disorders such as depression and anxiety, which are used by the population to confront stressors, and healthcare workers are not the exception. This was an analytical cross-sectional study, conducted from August to November 2022, via an online survey. Prevalence and severity of depression, anxiety, and stress were assessed via the DASS-21 test, and coping strategies were assessed via the CSSHW test. The sample consisted of 256 healthcare workers and of those, 133 (52%) were males with a mean age of 40.4 ± 10.35, and 123 (48%) were females with a mean age of 37.28 ± 9.33. Depression was prevalent in 43%, anxiety in 48%, and stress in 29.7%. Comorbidities were a significant risk factor for both depression and anxiety with an OR of 10.9 and 4.18, respectively. The psychiatric background was a risk factor for depression with an OR of 2.17, anxiety with an OR of 2.43, and stress with an OR of 3.58. The age difference was an important factor in the development of depression and anxiety. The maladaptive coping mechanism was prevalent in 90 subjects and was a risk factor for depression (OR of 2.94), anxiety (OR of 4.46) and stress (OR of 3.68). The resolution coping mechanism was a protective factor for depression (OR of 0.35), anxiety (OR of 0.22), and stress (OR of 0.52). This study shows that mental health disorders are highly prevalent among healthcare workers in Mexico and that coping strategies are associated with their prevalence. It also implies that not only occupations, age, and comorbidities might affect mental health, but also the way patients confront reality and the behavior and decisions they take towards stressors.

## 1. Introduction

Prior to COVID-19, mental health disorders such as anxiety and depression had a relatively high prevalence of around 13% of the population, however, after the pandemic, there was a clear increase in rates of anxiety being up to 50.9%, depression up to 48.3%, and stress up to 81.9% being mainly associated with infection rates and reductions in mobility [1,2,3,4].

The COVID-19 pandemic has had a severe impact on many aspects of society including the economy, education, preventive medicine, and of course, mental health [5,6]. Several studies have deepened the knowledge of mental health during the pandemic. Longitudinal ones showed that in most cases, the impact of the event had a significant reduction after the initial outbreak in 2020; however, the prevalence of PTSD symptoms was still above cut-off scores, and there were no reductions in anxiety, stress, and depression, thus stating that the levels of these remained constant [7,8,9].

Therefore, it can be asserted that some outcomes of the pandemic have been the increase in both the incidence and prevalence of mental health disorders such as depression, stress, anxiety, sleep problems, and posttraumatic symptoms being the highest in those who are healthcare workers or that have had relatives with COVID-19 [10,11,12,13].

On the other hand, healthcare workers have had to face many challenges during the pandemic, having to make almost impossible decisions with favorable or detrimental outcomes for their patients, being under persistent stress, and even developing a moral injury alongside proper mental health disorders such as depression or anxiety [14,15,16,17].

Having a global prevalence of mental health disorders of around 24% to 30% and, whether because of sociocultural differences, low-income, different approaches to death and disease, and hesitation towards vaccination, there has also been a clear disparity during the pandemic between developed and developing countries [18,19].

These mental health disorders and their incidence can also be affected by the way people cope with problems, challenges, or life itself [20,21]. There exist many coping models such as the coping circumplex model and stress-transactional coping [22,23,24,25,26].

The stress-transactional model of coping is one of the most known, utilized, and important models of coping, having been developed by Lazarus and Folkman in 1984. It is a macro and microanalytic approach model of coping in which they specify strategies to confront and cope with problems: confrontative coping, seeking social support, self-controlling, distancing, accepting responsibility, planful problem-solving, and positive reappraisal [24,25].

These coping strategies can also be classified into different types, such as resilience, maladaptive, resignation, and distancing, and have helped the population to regulate themselves and be resilient towards the pandemic [27,28,29,30,31].

The study hereby pretends to describe and explain the different types of coping strategies and their association with mental health disorders in healthcare workers during the COVID-19 pandemic in a state in Mexico.

The main objective of our study was to answer whether coping strategies might or might not affect the prevalence of mental health disorders. Moreover, the roles that gender, COVID-19 infection, death of relatives, comorbidities, and occupation play in mental health disorders were to be assessed.

## 2. Materials and Methods

### 2.1. Study Design and Data Sources

The present research was designed as an analytical cross-sectional study with a scope on mental health disorders such as depression, anxiety, and stress and the coping strategies utilized by health professionals in Tabasco, Mexico.

Data was collected via an online questionnaire designed to assess the prevalence of mental health disorders and coping strategies. The questionnaire was divided into four different groups of questions. The first one asked about informed consent, acknowledging that the data thereby answered and gathered was going to be used for research and academic purposes only. The second group focused on gathering information respective to personal data (age, gender, municipality, COVID-19 antecedents, psychiatric background, and the death of relatives due to COVID). The third group of questions was focused on the data gathering of mental health disorders, and finally, the fourth group focused on coping strategies.

### 2.2. Study Setting

This study was conducted between August and November of 2022, during the sixth COVID-19 wave, via an online questionnaire distributed to various health professionals independently of their adscription hospital or clinic. Recruitment started on 3 June 2022 and ended on 25 November 2022.

### 2.3. Participants and Procedure

Participants in the study were a convenience sampling consisting of volunteer targeted subjects, all surveyed had to accept informed consent.

The rationale behind convenience sampling was the fact that in Mexico and its states, there is no sufficient data available to the general population to assess the specific number of healthcare workers, therefore, being highly difficult to calculate a representative sample.

Participants were convened via promotion in medical faculties, hospitals, private clinics, and family medical units in Tabasco, Mexico.

Inclusion criteria: Both genders were admitted, ages from 18 to 70 years were accepted, only active healthcare workers were admitted, only healthcare workers inhabiting Tabasco were admitted, and all participants had to answer all sections of the questionnaire.

Exclusion criteria: Participants that did not accept informed consent or that did not answer all sections of the questionnaire.

The questionnaire was hosted online to collect data from all subjects. A total of 312 participants accessed the questionnaire, and 309 responded to the whole questionnaire, however, only 256 met the criteria and also gave informed consent.

### 2.4. Measures, Variables, and Data Collection

All subjects answered four groups of questions in the questionnaire, the first one required them to give informed consent, and the second focused on gathering information for epidemiological data (age, gender, municipality, COVID-19 antecedents, psychiatric background, and the death of relatives due to COVID).

To assess mental health disorders, the 21-item Depression, Anxiety, and Depression Scales (DASS-21 test) was utilized, consisting of a tripartite model of emotion, measured with a Likert-like scale (in which 0 was never, 1 sometimes, 2 many times, and 3 almost always). Low positive affectivity, negative affectivity, and psychophysiological agitation possessed both great internal consistency and reliability, having been validated in the Mexican population in the year 2006 with a Cronbach’s alpha value of 0.86 and 0.81, 0.76, and 0.79 for the subscales of depression, anxiety, and stress, respectively [32].

To assess the coping strategies used by the healthcare workers, the Coping Strategies Scale for Health Workers was utilized. The theoretical input utilized in Coping Strategies Scale for Health Workers is based on both Lazarus and Folkman and Omar’s works on the stress-transactional model [33,34,35]. This scale consists of 24 items and is answered in a Likert-like scale (never, sometimes, and always) giving, as a result, one or more coping strategies (maladaptive, resignation, resolution, and distancing coping strategies) utilized by the subject to confront problems or stress. This test has relatively good consistency and reliability having a Cronbach’s alpha value of 0.71 [35].

### 2.5. Statistical Analysis

Data were analyzed using Microsoft Excel (2021 version) and IBM SPSS Statistics 26.0. Qualitative variables were assessed via the Chi-square test and OR (odds ratio) was calculated. Chi-square was calculated via contingency tables comparing the main dependent variables: anxiety, depression and stress, and coping strategies with other qualitative variables such as gender, occupation, COVID-19 infection, and the death of relatives. Odds-ratio was calculated to assess protective and risk factors. Data imputation was to assess whether gender, comorbidities, psychiatric background, COVID-19 infection, death of relatives, and coping strategies were associated with more or less prevalence of depression, anxiety, and/or stress. The continuous variables were assessed for normality using the Kolmogorov-Smirnov test and they all showed a normal distribution (*p* < 0001). Due to this, a parametric inferential test was used: Student´s *t*-test between the variables previously mentioned to assess differences between group means. A *p*-value of 0.05 was considered statistically significant.

## 3. Results

A total of 256 subjects were studied, 133 (52%) were males and 123 (48%) were females. The mean age was 38.9 with a standard deviation of 9.98, showing normal distribution via the Kolmogorov–Smirnov test (*p* < 0.001) The mean age for males was 40.4 (SD of 10.35) and for females was 37.28 (SD of 9.33), having significant differences with a *p* of 0.012. Ten (3.9%) subjects, six women and four men were over 60 years.

One hundred and forty-one (55.1%) had no comorbidities while 115 (44.9%) had at least one comorbidity (among diabetes, hypertension, CKD, COPD, cardiovascular disease, and endocrinological disease), 199 (77.7%) had no psychiatric background, while 57 (22.3%) had a psychiatric background There were significant differences (*p* < 0.001) between the age of those with comorbidities and those with no comorbidities, the same applied with those with psychiatric background (*p* < 0.01).

Of the 256 subjects, 15 (5.9%) had an administrative occupation, 72 (28.1%) were medical doctors, 102 (39.8%) were nurses, 31 (12.1%) were technicians, and 36 (14.1%) were grouped as others.

Sixty-nine subjects (27%) had no history of SARS-CoV-2 infection, while 187 (73%) had infection confirmed by RT-PCR. Moreover, 167 (65.2%) had no relatives who died due to COVID-19 but 89 (34.8%) had at least one relative dead because of COVID-19 infection.

As for mental health disorders, depression had a prevalence of 110 (43%), of which 53 (48.18%) had mild depression, 34 (30.9%) had moderate depression, 19 (17.27%) had severe depression, and only 5 (4.54%) had extremely severe depression.

Anxiety was prevalent in 123 (48%) subjects, 75 (36.58%) had mild anxiety, 37 (30%) had moderate anxiety, 10 (8.13%) had severe anxiety, and only 1 (0.81%) had extremely severe anxiety.

Stress was prevalent in 76 (29.7%). Fifty-nine (77.63%) had mild stress, fourteen (18.42%) had moderate stress, only three (3.94%) had severe stress and no subject had extremely severe stress.

The mental health contingency table can be seen in Table 1 and distribution by occupation can be seen in Table 2.

Mean age difference proved to be significant via Student´s t-test in both depression (43.71 vs. 35.27, *p* < 0.001) and anxiety (41.81 vs. 36.2, *p* < 0.001) but was not significant in the case of stress (38.34 vs. 39.13, *p* = 0.56) (Figure 1, Figure 2 and Figure 3).

As for coping strategies, distancing coping was the most common with 133 (51.95%) subjects, 73 (54.88%) males and 60 females (45.11%) (*p* = 0.32). The second most common was resignation with 125 (48.82%) subjects, 75 (60%) males and 50 (40%) females (*p* = 0.12), followed by resolution coping mechanism, present in 108 (42.18%) subjects, 53 (49.07%) males and 55 (50.92%) females (*p* = 0.45). Finally, the least common coping mechanism was maladaptive coping, with 90 (35.15%) subjects, 55 (61.11%) males and 35 (38.88%) females (*p* = 0.03).

The distribution of coping strategies by occupation can be seen in Table 3.

Finally, of the 90 subjects with a maladaptive coping mechanism, 54 (59.99%) had depression, 64 (71.11%) had anxiety, and 43 (47.77%) had stress. Of the 108 subjects with resolution coping mechanisms, 31 (28.7%) had depression, 30 (27.77%) had anxiety, and 24 (22.22%) had stress. Of the 125 subjects with resignation coping, 61 (48.8%) had depression, 66 (52.8%) had anxiety, and 30 (24%) had stress. Of the 133 subjects with a distancing coping mechanism, 53 (39.84%) had depression, 65 (48.87%) had anxiety, and 43 (32.33%) had stress. The distribution of the coping strategies among the different mental health disorders is presented as follows (Table 4):

## 4. Discussion

The present study demonstrates that the prevalence of mental health disorders has a high prevalence among healthcare workers during the COVID-19 pandemic. Moreover, it is pointed out that the coexistence of comorbidities and psychiatric backgrounds may act as risk factors and that there is a clear relation between coping strategies and mental health disorders.

It is stated that the COVID-19 pandemic affected many aspects of society, including mental health and the existence of this perennial stressor only increased the already high prevalence and incidence of mental health disorders such as depression, anxiety, and stress [36,37,38,39,40].

Specifically, depression had a prevalence of 43%, anxiety of 48%, and stress of 29.7% which although not as high, is consistent with most studies, reviews, and meta-analyses on mental health disorders in healthcare workers in other countries [41,42,43,44].

This study also coincided with some studies made in Mexico in which it is stated that depression, stress, and anxiety are the main problem that affects more than 30% of healthcare personnel, therefore requiring them to be approached by institutions and mental health professionals [45,46,47,48].

Contrary to what is commonly believed, this study showed that at least, in our sample, the female gender had a lesser prevalence of depression even becoming a protective factor, but the stress was significantly higher. This might be derived from sociocultural differences between countries, though it requires more studies [49,50].

Gender differences in mental health are to be assessed, normally, the gap between males and females is wider and inclined towards females, which was not the case in our study [51,52].

Also, one of the pivotal factors in this work was to study whether different coping strategies were related to specific mental health disorders. Hereby it was found that maladaptive coping strategy was closely and significantly related to the development of depression, stress, and anxiety and that resolution coping strategy was a protection factor [53,54,55,56].

A non-intentional outcome of the study was the fact that although the mean age of our population was 38.9 years old, ten of our subjects had over sixty years which is not considered a common practice in other countries mainly because of the COVID-19 pandemic [57,58].

However, this is a relatively common practice in Mexico. Although the government convened all clinicians that presented comorbidities and were over 60 years to not engage in medical practice during the pandemic due to being an important risk factor for COVID-19 and death, ultimately it was not mandatory [59,60,61].

This study has several weaknesses that might be solved with changes in the design of further research. First, the sample was not calculated for convenience, due to the lack of data available about public and private healthcare, being an important limitation of the study so to lessen biases, a calculation of the sample might have a better and higher significance.

Also, the sample is not representative of Mexico nor it is intended to be. The sample as stated in the Methods was only from a state (Tabasco) in southeastern Mexico.

To address this weakness continuous communication with the health sector and the different institutions shall be kept essential throughout the entire study. Moreover, a much larger sample shall increase significance.

Secondary to what was stated in the prior paragraph, modifying the design from a cross-sectional to a longitudinal study might enlighten the understanding of mental health disorders for their high prevalence might be linked to the peaks of pandemics or changes in health policies in specific contexts.

A very important confounding factor is the fact that the classification of coping strategies was limited by the test used in only maladaptive, resolution, resignation, and distancing coping strategies, thus, it is worth noting that the coping strategies may be variable depending on the theoretical model utilized in other studies. Moreover, specific stressors were not part of the study, which may answer why certain occupations are more prone to certain mental health disorders. Moreover, current psychiatric treatment, non-drug therapies, admission to ICU during COVID-19, and current SARS-CoV-2 infection were not stated as variables that might change the landscape of risk and protection factors for mental health disorders.

We recommend that in further studies, a longitudinal analysis is stated, with a statistically larger sample and a calculated sample should be part of the study design. Other variables should be included, such as psychiatric treatment, non-drug therapies, the severity of COVID-19, admission to ICU during COVID-19 infection, and current SARS-CoV-2 infection.

## 5. Conclusions

This study denotes that during the COVID-19 pandemic healthcare workers have had a high prevalence of mental health disorders such as depression, anxiety, and stress. The development of these disorders is closely related to the use of coping strategies. Maladaptive coping strategies have been shown to be a main risk factor while resolution coping strategies have been shown to be a protective factor.

Therefore, it is implied that not only occupations, age, and comorbidities might affect mental health but also the emotional intelligence with which the patients confront reality and their behavior during and towards problems and stressors.

Also, it is notable that at least in this sample, health workers do not have sufficient emotional tools which might be the reason why mental health disorders are so prevalent.

To decrease both the incidence and prevalence of mental health disorders in healthcare workers, it might be essential for private and public health institutions to have protocols and programs in which they prevent, diagnose, treat, and accompany their employees.

## Figures and Tables

**Figure 1 ijerph-20-04230-f001:**
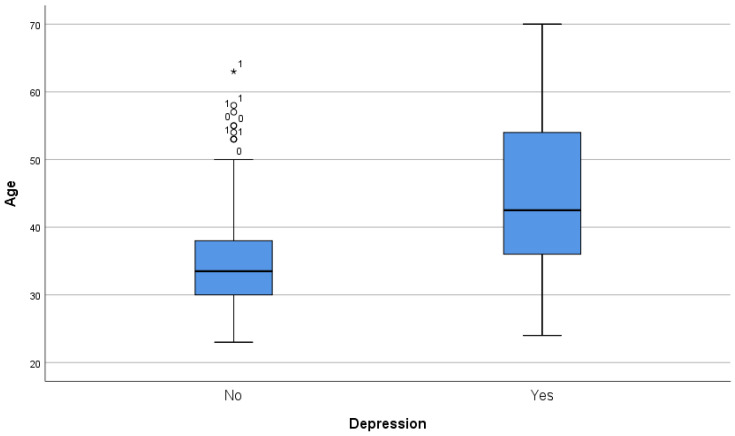
Box and whiskers graph, age, and depression diagnosis. ^1^ women, ^0^ men, * *p* < 0.05.

**Figure 2 ijerph-20-04230-f002:**
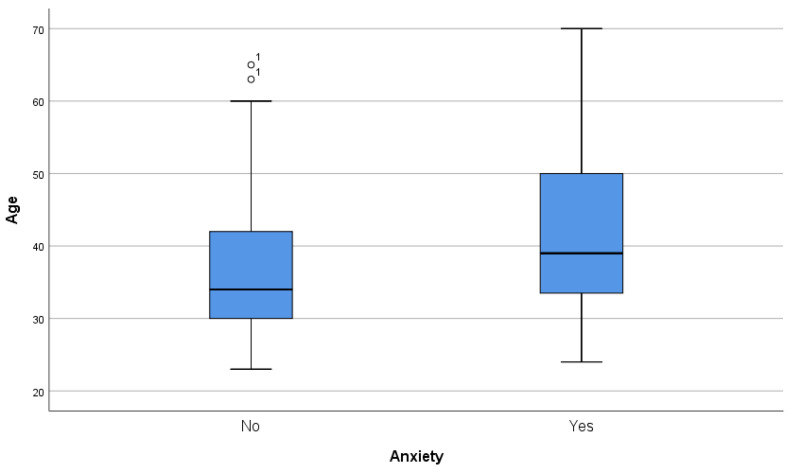
Box and whiskers graph, age, and anxiety diagnosis. ^1^ women.

**Figure 3 ijerph-20-04230-f003:**
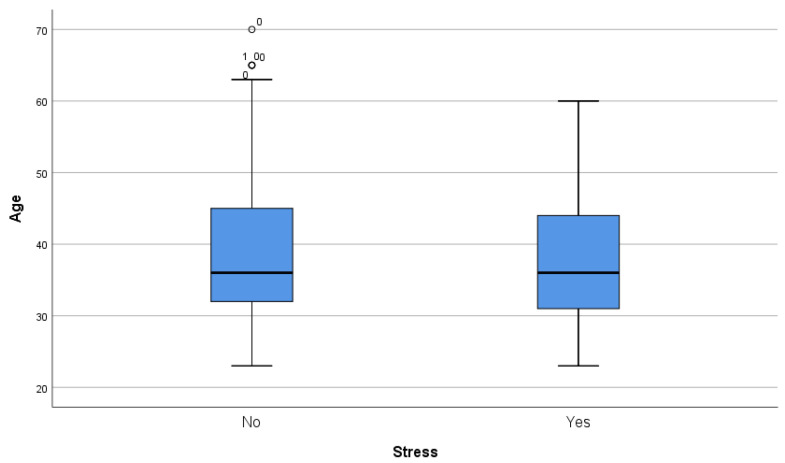
Box and whiskers graph, age, and stress diagnosis. ^1^ women, ^0^ men.

**Table 1 ijerph-20-04230-t001:** Contingency table, factors associated with depression, anxiety, and/or stress.

Variable	Depression		Anxiety		Stress	
	Chi-Square Test	OR (95% CI)	Chi-Square Test	OR (95% CI)	Chi-Square Test	OR (95% CI)
Gender	0.047 *	0.6 (.36 to 0.99)	0.2	0.72 (0.44 to 1.18)	0.04 *	1.78 (1.02 to 3.02)
Comorbidities	<0.001 **	10.9 (6.1 to 19.65)	<0.001 **	4.18 (2.47 to 7)	0.4	1.24 (0.72 to 2.12)
Psychiatric background	0.01 *	2.17 (1.19 to 3.96)	0.004 **	2.43 (1.06 to 1.39)	<0.001 **	3.58 (1.6 to 8)
COVID-19 infection	0.45	1.24 (0.7 to 2.18)	0.49	1.78 (1.01 to 3.13)	0.009 **	1.25 (1.03 to 1.52)
Death of relatives	0.16	0.68 (0.4 to 1.1)	0.1	1.54 (0.91 to 2.58)	0.059	1.69 (0.97 to 2.95)

* *p* < 0.05; ** *p* < 0.005.

**Table 2 ijerph-20-04230-t002:** Distribution of mental health disorders by occupation.

Occupation	Depression			Anxiety			Stress		
	Yes (*n*,%)	No (*n*, %)	Chi-Square Test	Yes (*n*,%)	No (*n*, %)	Chi-Square Test	Yes (*n*,%)	No (*n*, %)	Chi-Square Test
Administrative	7, 46.6	8, 53.33	*p* = 0.003 **	10, 66.66	5, 33,33	*p* = 0.01 *	4, 26.66%	11, 73.33	0.13
Medical doctor	39, 54.16	33, 45.83		38, 52.77	34, 47.22		15, 20.83	57, 51.38	
Nurse	42, 41.17	60, 58.82		47, 46.07	55, 53.92		35, 34.31	67, 65.68	
Technician	4, 12.9	27, 87.09		7, 22.58	24, 77.41		7, 22.58	24, 77.41	
Others	18, 50	18, 50		21, 58.33	15, 41.66		15, 41.66	21, 58.33	

* *p* < 0.05; ** *p* < 0.005.

**Table 3 ijerph-20-04230-t003:** Distribution of coping strategies by occupation.

Occupation	Maladaptive Coping			Resolution Coping			Resignation Coping			Distancing Coping		
	Yes (*n*,%)	No (*n*, %)	Chi-Square Test	Yes (*n*,%)	No (*n*, %)	Chi-Square Test	Yes (*n*,%)	No (*n*, %)	Chi-Square Test	Yes (*n*,%)	No (*n*, %)	Chi-Square Test
Administrative	7, 46.6	8, 53.33	*p* = 0.01 *	6, 40	9, 60	*p* = 0.35	7, 46.6	8, 53.33	*p* = 0.03 *	3, 20	12, 80	*p* < 0.001 **
Medical doctor	26, 36.1	46, 63.88		34, 47.22	38, 52.77		41, 56.94	31, 43.05		29, 40.27	43, 59.72	
Nurse	41, 40.19	61, 59.8		38, 37.25	64, 62.74		55, 53.92	47, 46.07		51, 50	51, 50	
Technician	3, 9.67	28, 90.32		17, 54.83	14, 45.16		12, 38.7	19, 61.29		24, 77.41	7, 22.58	
Others	12, 33.33	24, 66.66		13, 36.11	23, 63.88		10, 27.77	26, 72.22		26, 72.22	10, 27.77	

* *p* < 0.05; ** *p* < 0.005.

**Table 4 ijerph-20-04230-t004:** Coping strategies and mental health disorders.

Coping Strategies	Depression		Anxiety		Stress	
	Chi-Square Test	OR (95% CI)	Chi-Square Test	OR (95% CI)	Chi-Square Test	OR (95% CI)
Maladaptive	*p* < 0.001 **	2.94 (1.73 to 5)	*p* < 0.001 **	4.46 (2.56 to 7.78)	*p* < 0.001 **	3.68 (2.1 to 6.47)
Resolution	*p* < 0.001 **	0.35 (0.2 to 0.59)	*p* < 0.001 **	0.22 (0.13 to 0.38)	*p* = 0.02 *	0.52 (0.3 to 0.92)
Resignation	*p* = 0.06	1.59 (0.96 to 2.62)	*p* = 0.13	1.45 (0.88 to 2.37)	*p* = 0.052	0.58 (0.33 to 1)
Distancing	*p* = 0.29	0.76 (0.46 to 1.26)	*p* = 0.78	1.07 (0.65 to 1.75)	*p* = 0.33	1.3 (0.76 to 2.23)

* *p* < 0.05; ** *p* < 0.005.

## Data Availability

The datasets used and/or analyzed in the current study are available from the corresponding author upon reasonable request.

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
