# Peer review of "Mental Health Disorders and Coping Strategies in Healthcare Workers during the COVID-19 Pandemic: An Analytical Cross-Sectional Study in Southeastern Mexico"

_ijerph, 2023, doi:10.3390/ijerph20054230_

Round 1

Reviewer 1 Report

Thank you for the opportunity to review this study entitled “Mental health disorders and coping strategies in healthcare workers during the COVID-19 pandemic: An analytical cross-sectional study in Mexico” (ijerph-2215753).

The research investigated the psychological effect of the COVID-19 pandemic, by focusing on the severity of depression, anxiety and stress, as well as on the use of coping strategies. The research involved a sample of 256 healthcare workers.

In my opinion, the research topic is relevant, and the study is interesting. Parallelly, some issues need to be addressed before the paper will be suitable for publication.

·       Abstract: the information about the samples should be deepened (Mean age and SD? Percentage of men and women?) to provide a clear picture of what will be presented in the paper.

·       Abstract: please avoid reporting all these indices in the abstract (e.g., p<0.005). To avoid making it confusing and difficult to read, none or only the main indexes should be reported.

·       Introduction: this section requires more work to provide an overview of the mental health effects of the pandemic, to provide a theoretical framework for the entire study. In my opinion, it would also be good to refer to trend or longitudinal studies, if any, to enrich this section. Since the authors frame this study considering the impact that COVID-19 has on a psychological level, I suggest some research to propose a comprehensive framework in the introduction, which should be supplemented with further literature search by the authors:

o   Hyland et al., 2021; doi: 10.1016/j.psychres.2021.113905.

o   Gori & Topino, 2021; doi: 10.3390/ijerph18115651

o   Wang et al., 2020; doi: 10.1016/j.bbi.2020.04.028

To find the suggested articles, the authors can use this source: https://www.doi.org/

·       Method: Please, calculate the Cronbach's alpha values for each measure in the present sample.

·       The “Conclusions” sections should be enriched by highlighting the practical implication of this research.

Best wishes

Author Response

file with observations attached

Reviewer 2 Report

Dear authors, 

Thanks for presenting me with the opportunity to review this interesting work. I have some queries :

1) Liker scale should be corrected to Likert scale ( Line 108)

2) The sentence ' random, non -probability, purposive sampling' ( Lines 86-87) is redundant and should be replaced with convenience sampling.

3) It is interesting that the inclusion age group 18-70 years old is used in this study. Perhaps the authors could elaborate on the unique nature of the study population whereby older healthcare workers are employed. This practice is commendable and is not commonly seen in certain populations, and thus is interesting to explore as well. 

4) In the discussion section, it is mentioned that the female gender had lesser prevalence of depression. It will be beneficial at this juncture to contrast this data with other studies done globally scrutinizing gender roles in depression during the pandemic :

Francis, B., Ken, C. S., Han, N. Y., Ariffin, M. A. A., Yusuf, M. H. M., Wen, L. J., ... & Said, M. A. (2021). Religious coping during the COVID-19 pandemic: Gender, occupational and socio-economic perspectives among Malaysian frontline healthcare workers. Alpha Psychiatry22(4), 194.

Author Response

file with observations attached

Reviewer 3 Report

Thank you very much for your study. In the theoretical perspective of coping, its authors argue that stressful events must be understood contextually, considering the dynamic and changeable transactions between individuals and the environment. And placing the study in the perspective of COVID-19 is interesting. However, the data is poorly explained; there is no sample calculation, generalization parameters or a sustained discussion.

Below I explain my main concerns.

1. Working Mental health disorders and coping is very broad and much has already been produced about it. What is the theoretical model of coping that you work with? This needs to be clear;

2. The introduction is incomplete, it does not bring a series of new and interesting texts to understand the problem. The data in your introduction must be updated, present the prevalence of the phenomenon and the most recent data possible.

It must also necessarily POINT OUT AND PRESENT the knowledge gap that it came to fill, that is, what one needs to know about the subject and justify it. Originality (HIGHLY RECOMMENDED). Remember that the gap must be auditable, so it is necessary to carry out a systematic search or point out a current literature review (integrative or systematic);

3. Recruitment is not clear, neither are inclusion criteria and data collection. Where did you get the participants?

How did the amount increase?

Who was the initial population?

How many accessed the instrument?

How many responded? How many refused?

Who was the source population? How many?

Where is the sample calculation? What is the power of this sample?

Where was the form hosted?

What are the advantages and disadvantages of the form where it was hosted?

How was the generalization of the sample guaranteed?

4. How were the ORs calculated? Where? How was the data imputation?

5. The study makes generalizations throughout the discussion that its data do not support. For example, pointing out that “The present study demonstrates that the prevalence of mental health disorders has a high prevalence among healthcare workers during the COVID-19 pandemic, being primarily affected by the coexistence of comorbidities and psychiatric backgrounds”, cannot be said without a theoretical background that points out how you analyzed the variables and without sample data.

Your work is not representative of the Mexico Healthcare workers population.

Author Response

file with observations attached

Reviewer 4 Report

The study entitled “Mental Health Disorders and Coping Strategies in Healthcare 2 Workers During the Covid-19 Pandemic: An Analytical Cross-Sectional Study in Mexico” is an interesting and potentially valuable addition to the literature on the prevalence of psychological disorders in a selected sample of healthcare workers. The study rests on the likely readers’ interest and concern regarding the impact of the COVID-19 pandemic on people who have been on the frontline of every country’s response to the pandemic. The manuscript deserves publication after a few concerns are successfully addressed:

The abstract needs to be rewritten to clarify content as well as fix grammatical issues and typos. Most importantly, the abstract is intended to entice readers to examine the entire manuscript. Thus, it should be written in a narrative format that highlights the general features and value of the authors’ study. For instance, p values and other statistical artifacts should be left to the result section.

In the first sentence of the abstract, it is important to make it clear that the COVID-19 pandemic is the context and that mental disorders are the phenomena that may take place in such a context. The last sentence may briefly cover the practical and theoretical implications of the study.

In the introductory section, it is also essential to make it clear that the COVID-19 pandemic is the environmental context for the phenomena that the authors wish to explore. What are the specific research questions that the authors have selected for their research? What are the hypotheses that the authors wish to test? What is the theoretical model upon which research questions and hypotheses rest?

 The introduction must be followed by a thorough literature review of the particular mental disorders people are likely to have experienced during the COVID-19 pandemic. An extensive review of coping strategies is also necessary.  

 The method section needs more details about the sampling procedure adopted by the authors to ensure that the study can be replicated. What was the participation rate? What is the author’s rationale for using the selected assessment tools? If the authors want to assess the prevalence of different forms of distress, why did they rely only on the Depression, Anxiety, and Stress Scale (i.e., DASS-21)? If this is so, should the title be changed? Who are the authors of this scale? Has this scale been used in other contexts? The same questions apply to the Coping Strategies Scale for Health Workers.

 The data analysis section needs to be preceded by a section highlighting each of the choices of inferential statistics made and their respective rationale.

The discussion section does not adequately link the study's findings with those of the extant literature. The practical and theoretical implications of the authors’ study need to be addressed more thoroughly. Most importantly, the limitations of the study may benefit from more extensive coverage.

Author Response

file with observations attached

Round 2

Reviewer 4 Report

The authors of the paper entitled “Mental Health Disorders and Coping Strategies in Healthcare Workers During the Covid-19 Pandemic: An Analytical Cross-Sectional Study in southeastern Mexico” have made extensive revisions to the original document. Revisions have addressed most of the concerns raised in the earlier review. In my modest opinion, the paper deserves to be published.   

Author Response

We appreciate the comments made, which undoubtedly substantially improve the manuscript. We attach a general final review of the manuscript in writing and improvement of the methodology section.
Thank you and we remain attentive to the opinion of the manuscript.